# Optimizing Client Participation in Communication-Constrained Federated LLM Adaptation with LoRA

**DOI:** 10.3390/s25216538

**Published:** 2025-10-23

**Authors:** Faranaksadat Solat, Joohyung Lee

**Affiliations:** Department of Computing, Gachon University, Seongnam 13120, Republic of Korea; faranak1995@gachon.ac.kr

**Keywords:** federated learning, large language models, communication efficiency, client selection, parameter-efficient tuning

## Abstract

Federated learning (FL) enables privacy-preserving adaptation of large language models (LLMs) across distributed clients. However, deploying FL in edge environments remains challenging because of the high communication overhead of full-model updates. Recent advances in parameter-efficient fine-tuning (PEFT), particularly low-rank adaptation (LoRA), have substantially reduced update sizes by injecting lightweight trainable matrices into pretrained transformers, thereby making FL with LLMs more feasible. In this paper, we propose LoRaC-GA, a communication-aware optimization framework that dynamically determines the optimal number of clients to participate in each round under a fixed bandwidth constraint. We formulated a max-min objective to jointly maximize the model accuracy and communication efficiency and solved the resulting non-convex problem using a genetic algorithm (GA). To further reduce the overhead, we integrated a structured peer-to-peer collaboration protocol with $\log_{2}K$ complexity, enabling scalable communication without full connectivity. The simulation results demonstrate that LoRaC-GA adaptively selects the optimal client count, achieving competitive accuracy while significantly reducing the communication cost. The proposed framework is well-suited for bandwidth-constrained edge deployments involving large-scale LLMs.

## 1. Introduction

Federated learning (FL) has emerged as a promising paradigm for training machine learning models across distributed clients while preserving user privacy. Recently, attention has turned to adapting large language models (LLMs) in FL settings, enabling personalized natural language processing on edge devices without exposing sensitive data to the server. However, transmitting large-model updates in each round introduces a significant communication overhead, particularly in bandwidth-constrained wireless environments.

To address this, parameter-efficient fine-tuning (PEFT) techniques, such as low-rank Adaptation (LoRA), have been proposed. LoRA injects small, trainable low-rank matrices into pretrained transformer architectures while freezing the backbone, reducing the number of trainable parameters and the update size by orders of magnitude. This makes LoRA particularly suitable for the federated adaptation of LLMs in practical resource-constrained environments. Yet, a key question remains: *How many clients should participate in each round to balance model performance and communication cost under bandwidth constraints?*

Existing research on communication-efficient FL has explored various aspects of this tradeoff. Traditional client selection strategies rely on static [1] or randomized [2] sampling without accounting for the interplay between the client count and bandwidth budget. More recent work has considered adaptive selection under resource constraints [3,4], but the emphasis has been on convergence or fairness, not communication efficiency in LLM adaptation. LoRA-based tuning for FL has been introduced [5], but it assumes a fixed client count, leaving the cost-performance tradeoff unaddressed. Decentralized frameworks, such as ProxyFL [6], reduce server reliance through peer-to-peer (P2P) update sharing but assume full mesh connectivity, which is often impractical.

In contrast, we propose a bandwidth-aware optimization framework for LoRA-based federated LLM tuning. Our approach dynamically determines the number of participating clients per round while integrating a structured $\log_{2}K$ peer collaboration protocol to reduce communication complexity without sacrificing accuracy.

In summary, the main contributions of this study are as follows:The proposed framework enables communication-aware client selection for federated adaptation of LLMs using LoRA, specifically designed for constrained wireless environments.To capture the tradeoff between performance and cost, the problem is formulated as a max-min optimization that jointly maximizes model accuracy while minimizing communication overhead.To efficiently solve the resulting non-convex problem, a genetic-algorithm (GA)-based method is introduced, allowing dynamic determination of the optimal number of clients per round.In addition, a structured peer-to-peer collaboration protocol with $\log_{2}K$ complexity is incorporated, supporting scalable update sharing without requiring full connectivity.Finally, extensive simulations under varying bandwidth budgets confirmed that the framework maintains competitive accuracy while substantially reducing communication cost compared to baselines.

**High-Level Architecture.** Our solution comprises three layers: (i) *server layer*—an aggregator/scheduler that holds the frozen pretrained backbone $W_{0}$ and the global LoRA adapters $\theta_{t}$ and runs LoRaC-GA to choose the client count *K* each round under the budget *B*; (ii) *client layer*—the selected *K* clients, each with a private dataset $D_{i}$, that train only low-rank adapters (A,B) while keeping $W_{0}$ fixed (optionally, clients can engage in a sparse peer-to-peer overlay with per-node degree O(logK) to refine updates); and (iii) *communication layer*—downlink of $\theta_{t}$ and uplink of adapter deltas $\Delta \theta_{i}$ with per-client payload *S*, enforced by the budget *B*. In each round, *(schedule)* LoRaC-GA selects *K*, *(distribute)* the server sends $\theta_{t}$ (the frozen $W_{0}$ is not transmitted), *(local LoRA training)* clients update (A,B) on $D_{i}$, *(optional P2P)* clients exchange/average adapter deltas over the sparse overlay, *(aggregate)* clients upload $\Delta \theta_{i}$, and the server produces $\theta_{t+1}$ via FedAvg-style aggregation. This end-to-end flow is illustrated in Figure 1 and summarized procedurally in Algorithm 1.

The remainder of this paper is organized as follows. Section 3 introduces the system model and problem formulations. Section 4 presents the proposed LoRaC-GA framework. Section 5 provides performance evaluation, and Section 6 concludes the article.
**Algorithm** **1** LoRaC-GA: GA for client selection in federated LLM adaptation**Input:** Kmax, *B*, *S*, *R*, population size *P*, generations *G*, crossover rate pc, mutation rate pm, elitism ratio ρ**Output:** Optimal number of clients K*
1: Initialize population P(0)={K1,…,KP} with random integers in [1,Kmax]2: **for** g=1 to *G*
**do**3:     **for** each Ki∈P(g)
**do**4:         Compute C(Ki)=R·Ki·S5:         Estimate A(Ki) via simulation or profiling6:         Evaluate f(Ki)=minA(Ki),BC(Ki)7:     **end for**8:     Select parents using tournament selection9:     Apply crossover and mutation to generate new candidates10:    Form new population P(g+1) by combining elites and offspring11: **end for**12: K*←argmaxK∈P(G)f(K)13: **return** 
K*


## 2. Related Work

**Context and Terminology.** We study the federated *adaptation* of LLMs using PEFT, specifically LoRA. A central server coordinates many edge clients. The pretrained LLM backbone is kept frozen; only low-rank adapter parameters are trained and exchanged. Communication is constrained by a bandwidth budget *B*. The decision variable is the number of participating clients per round *K*: increasing *K* can improve accuracy (via data diversity) but raises communication cost. We formalize this accuracy–communication tradeoff in Section 3 and Section 4 and solve it with a GA. This primer orients non-specialist readers before we survey related strands.

### 2.1. Communication-Efficient FL and Client Selection

The seminal work FedAvg [1] introduced a simple yet effective algorithm for federated optimization, demonstrating how local training and model averaging can drastically reduce communication compared with centralized learning (CL). However, FedAvg struggles with heterogeneous data distributions and resource-constrained devices. To alleviate such bottlenecks, FedCS [2] accounts for device and network heterogeneity when selecting clients. TiFL [7] further improves performance by grouping clients with similar training speeds, achieving up to 6× faster convergence. Oort [8] employs guided client selection based on utility and system characteristics, demonstrating up to 14× faster time to accuracy. Recently, RBPS [9] introduced a reward-based payoff strategy in which clients make participation decisions based on accuracy, energy, and privacy tradeoffs. These studies highlight the critical role of client selection in improving FL efficiency although they primarily focus on *which* clients should participate rather than *how many* clients should join under a strict communication budget.

*Takeaway.* Prior client-selection works primarily decide *which clients* to involve; they do not optimize *how many* clients to select under an explicit bandwidth budget *B*.

### 2.2. Parameter-Efficient Fine-Tuning in FL and NLP

To reduce the prohibitive communication costs of large-scale models, PEFT has emerged as a promising approach. FedAdapter [10] integrates adapter modules into transformer-based models and achieves up to 155× faster convergence than do standard FL approaches. FedPrompt [11] enables federated prompt tuning by sharing soft prompt embeddings, reducing communication to only 0.01% of model parameters with minimal accuracy loss. FedPepTAO [12] incorporates adaptive optimization to mitigate the effects of non-IID client data in prompt tuning. FedLoRA [5] brings LoRA into FL but assumes a fixed client count per round, leaving the bandwidth-constrained cost–performance tradeoff unaddressed. *Takeaway.* PEFT/LoRA drastically reduces per-client payloads, but existing FL adaptations typically assume a fixed *K* and therefore do not address accuracy–communication co-optimization.

### 2.3. Generative and Multi-Criteria Client Selection

Recent research has explored sophisticated strategies for client selection. FedGCS [13] introduced a generative framework that encodes and decodes client selection decisions, enabling scalable and flexible selection policies. FedPROM [14] formulates client selection as a multi-criteria optimization problem, balancing model accuracy, latency, and resource usage. Although these approaches improve decision-making in client selection, they do not address parameter-efficient LLM adaptation or explicitly optimize the number of clients within bandwidth constraints.

*Takeaway.* Generative/multi-criteria policies enrich *how* clients are chosen, yet they do not target LoRA-based LLM adaptation with an explicit optimization of *K* under *B*.

### 2.4. GA-Based Optimization in FL

GA has been widely applied to solve non-convex optimization problems in communication and network systems owing to its ability to efficiently explore large solution spaces [15,16]. In FL, GA has been particularly useful for optimizing resource allocation and grouping decisions under heterogeneous environments. For example, clustered FL schemes, such as FedGM [17], apply GA to minimize both the average idle time and the group-creation cost across MEC servers, thereby improving convergence speed while maintaining accuracy. *Takeaway.* GA is well-suited for discrete, non-convex decisions in FL; here we use it to select the *client count K* that balances accuracy and communication.

**GA primer (non-specialist).** In our setting the chromosome encodes the integer client count *K*. The fitness is $f\left(K\right)=\min\;\left(A(K),\;B/(RKS)\right)$ with feasibility $1\;\leq \;K\;\leq \;K_{\max}$ and C(K)=RKS≤B. We use tournament selection, single-point crossover (probability $p_{c}$), integer mutation to another feasible *K* (probability $p_{m}$), and elitism ratio ρ. A typical configuration is population P∈[20,50] and generations G∈[20,50], $p_{c}\;\in \;\left[0.7,0.9\right]$, $p_{m}\;\in \;\left[0.05,0.15\right]$, and ρ∈[0.05,0.1]. The computational burden is dominated by evaluating A(K); since A(K) is profiled offline from training curves, the online GA search is lightweight (about P×G fitness calls). GA is robust to discrete, non-convex objectives but is stochastic; we fix the random seed for reproducibility.

**Other engines at a glance.** For reader orientation, we briefly summarize the referenced methods—beyond acronyms—with their focus, advantages, and limitations relative to our objective (bandwidth-aware optimization of the *number* of clients *K*). A comparative overview of these methods is provided in Table 1.

In summary, prior research has primarily optimized either (i) *which* clients participate in each round (e.g., FedCS, TiFL, Oort, RBPS, FedGCS, FedPROM) or (ii) *how to reduce per-client communication* via PEFT (e.g., FedAdapter, FedPrompt, FedPepTAO). The proposed framework, LoRaC-GA, is the first to explicitly optimize the *number of participating clients* under a fixed bandwidth budget for LoRA-based LLM adaptation. By combining a GA with structured communication overlays of complexity O(KlogK), the method effectively balances accuracy, efficiency, and scalability in bandwidth-constrained federated LLM training.

**Positioning and novelty.** Unlike prior PEFT/LoRA-based FL methods (e.g., FedAdapter, FedPrompt, FedPepTAO, FedLoRA) that typically assume a fixed number of participating clients, our framework *optimizes* the client count *K* under an explicit bandwidth budget via a max–min objective coupled with a GA-driven search. In addition, we integrate a structured P2P collaboration protocol with per-node degree O(logK) (overall O(KlogK) messaging), enabling scalable communication without full connectivity. This explicitly addresses the joint accuracy–communication trade-off that earlier work has not handled.

## 3. System Model

We consider a standard cross-device FL scenario consisting of a central server and $K_{\max}$ distributed clients. Each client possesses a private local dataset and participates in training a shared LLMs using LoRA for PEFT. Communication occurs over bandwidth-limited wireless links, and only a subset of clients is selected to participate in each training round.

**Backbone and adapters.** We adopt PEFT with LoRA, keeping the pretrained LLM backbone $W_{0}$*frozen* (no gradients; not transmitted) and optimizing only low-rank adapters. For a linear layer with weight *W*, we write $W=W_{0}+\Delta W$ with ΔW=AB, where $A\;\in \;R^{d\times r}$ and $B\;\in \;R^{r\times d}$, r≪d. During local training, backpropagation updates only (A,B), which substantially reduces computation and communication compared with updating the full backbone.

**Workflow.** As shown in Figure 1, at round *t* the server maintains $\left\{W_{0},\theta_{t}\right\}$, where $W_{0}$ is the frozen backbone, and $\theta_{t}$ is the current global LoRA adapter. Under the bandwidth budget, the server selects a subset of clients and transmits only $\theta_{t}$; the frozen backbone $W_{0}$ is *not* sent. Each selected client trains its local adapters $\theta_{i}$ on private data, while $W_{0}$ remains fixed; optionally, clients perform a lightweight peer-to-peer exchange over a structured overlay with per-node degree O(logK) (overall O(KlogK) messages) to refine updates. Clients then upload their adapter deltas $\Delta \theta_{i}$ (per-client uplink payload equals *S* per round); the server aggregates them in a FedAvg-style manner to obtain $\theta_{t+1}$, and the process repeats until a stopping criterion (e.g., *R* rounds or convergence).

### 3.1. Communication and Model Update

Let *R* denote the total number of communication rounds and *S* the size (in megabytes) of the LoRA adapter update transmitted by each client per round. Since LoRA updates are significantly smaller than are full model weights, they enable lightweight communication. For a given number of participating clients $K\leq K_{\max}$, the total communication cost is as follows:(1)C(K)=R·K·S.

Let *B* represent the maximum allowable communication budget. Any feasible client selection strategy must satisfy the constraint C(K)≤B.

### 3.2. Accuracy–Communication Tradeoff

Let A(K) denote the final model accuracy when *K* clients participate in each round. While A(K) generally increases with *K* due to improved data diversity, it often saturates or exhibits diminishing returns. Meanwhile, the communication cost increases linearly with *K*, giving rise to a fundamental tradeoff between the accuracy and communication efficiency.

### 3.3. Optimization Problem

To balance the accuracy and bandwidth usage, we formulate a max-min optimization problem that jointly considers the model performance and communication cost as follows:$$\max_{K\in \{1,2,\ldots ,K_{\max}\}}\min\left(A\left(K\right),\frac{B}{C(K)}\right),$$
where A(K) represents the empirical model accuracy under *K* participating clients, and B/C(K) is introduced as a measure of communication efficiency. In this formulation, C(K) represents the total communication cost when *K* clients are selected, and *B* is the bandwidth budget. The ratio B/C(K) therefore expresses how efficiently the available bandwidth is utilized: values close to 1 indicate that the system operates well within the budget (high efficiency), whereas values approaching 0 imply that communication demand is high relative to the budget (i.e., low efficiency). This normalized representation provides a comparable scale between accuracy and communication usage, enabling a balanced optimization. Similar efficiency ratios have been employed in prior work on resource-aware FL [18,19].

These are subject to the follows constraints: (2)$$\begin{array}{cc} \; & 1\leq K\leq K_{\max},\;K\in Z, \end{array}$$(3)$$\begin{array}{cc} \; & C(K)=R\cdot K\cdot S\leq B. \end{array}$$

This optimization problem is inherently non-convex due to the empirical and non-differentiable nature of A(K), which is typically obtained via simulation or profiling. Furthermore, the discreteness of *K* prevents the use of gradient-based techniques. Similar non-convex formulations have been discussed in the FL literature [20,21], and meta-heuristic methods such as GA are widely recognized for addressing such discrete optimization problems [17,22]. Therefore, we adopted a metaheuristic approach using a GA, which is well-suited for black-box optimization over discrete, non-convex domains. The GA efficiently searches the feasible space to identify the optimal number of clients that jointly maximizes the model accuracy and communication efficiency.

### 3.4. Client Collaboration Protocol

To further reduce the communication overhead, we incorporate a structured peer-to-peer (P2P) collaboration scheme inspired by ProxyFL [6]. Rather than relying on fully connected P2P communication, each client exchanges updates with a subset of peers following a $\log_{2}K$ topology per round. This reduces the number of P2P transmissions from $O(K^{2})$ to O(KlogK).

Each client aggregates the received LoRA updates using proxy-based averaging or consensus. This structured, scalable message-passing mechanism supports decentralized collaboration while avoiding bottlenecks, making it especially effective for bandwidth-constrained edge deployments.

## 4. Proposed Method: LoRaC-GA

In this section, we present *low-rank adaptation-based client selection using GA (LoRaC-GA)*—a framework designed to optimize the number of participating clients in communication-constrained federated LLM adaptation. The core idea is to identify the optimal client count *K* that maximizes the model accuracy and communication efficiency under a fixed bandwidth budget.

**Overview.** Building on the high-level architecture in Section 1 and the system model in Section 3, this section details the following: (i) the client-collaboration protocol, (ii) the problem-specific formulation, and (iii) the GA components used to select *K* under a fixed budget *B*. Concretely, given *B* and per-client payload *S*, we select K∈K to maximize $f\left(K\right)=\min\;\left(A(K),\;B/(RKS)\right)$; A(K) is profiled empirically, and a GA efficiently searches the discrete feasible space.

### 4.1. Client-Collaboration Protocol

To reduce message complexity without full connectivity, we adopt a structured peer-to-peer (P2P) overlay in which each client exchanges adapter updates with O(logK) peers (overall O(KlogK) messages). The pretrained backbone remains frozen; only LoRA adapters are communicated. Clients optionally average exchanged deltas locally before uplink, which improves update quality under bandwidth constraints and integrates seamlessly with the GA-driven client-count selection.

### 4.2. Problem-Specific Formulation

We restate the optimization problem from Section 3 as$$\max_{K\in \{1,2,\ldots ,K_{\max}\}}\min\left(A\left(K\right),\frac{B}{C(K)}\right),$$
where the following holds:A(K) denotes the model accuracy achieved when *K* clients participate per round.C(K)=R·K·S is the total communication cost.*B* is the communication budget.

The ratio B/C(K) reflects the communication efficiency, whereas the pointwise minimum captures the tradeoff between performance and cost. The feasible domain is defined as follows:$$K=\{K\in Z|1\leq K\leq K_{\max},\;C\left(K\right)\leq B\}.$$

Due to the non-convex, non-differentiable nature of A(K) and the discreteness of *K*, we adopted a GA to solve this black-box optimization problem.

### 4.3. GA Framework

We employ a GA to solve the above max-min optimization, leveraging its effectiveness for discrete, non-convex optimization tasks in FL [17]. Each individual in the population encodes a candidate value of *K*, and the fitness function is defined as follows:$$f\left(K\right)=\min\left(A\left(K\right),\frac{B}{R\cdot K\cdot S}\right).$$

The GA evolves a population over a fixed number of generations using the following operators:

**Selection:** Tournament selection is used to compare and select parents:$$\mathrm{Select}\left(K_{i},K_{j}\right)=\arg\max\left\{f\left(K_{i}\right),f\left(K_{j}\right)\right\}.$$

**Crossover:** With crossover probability $p_{c}$, two parents generate an offspring $K^{\prime }$ as follows:$$K^{\prime }=\left\{\begin{array}{cc} K_{i}, & \mathrm{with}\;\mathrm{probability}\;0.5,\\ K_{j}, & \mathrm{otherwise}. \end{array}\right.$$

**Mutation:** With mutation probability $p_{m}$, a candidate $K^{\prime }$ mutates to the following:$$K_{\mathrm{mut}}=\left\{\begin{array}{cc} \mathrm{rand}(1,K_{\max}), & \mathrm{with}\;\mathrm{probability}\;p_{m},\\ K^{\prime }, & \mathrm{otherwise}. \end{array}\right.$$

**Elitism:** A fraction ρ of top-performing individuals is retained unchanged to preserve the best solutions.

This process is repeated over *G* generations. The optimal client count is selected as follows:$$K^{*}=\arg\max_{K\in P^{\left(G\right)}}f\left(K\right),$$
where $P^{\left(G\right)}$ is the final population.

### 4.4. Theoretical Insight

To justify the use of a metaheuristic solver, we characterize the structure of the objective as follows:

**Lemma** **1.**
*Let A(K) be a bounded, non-monotonic, and non-convex function representing model accuracy, and let C(K) be a strictly increasing linear function. Then the objective*

$$\max_{K}\min\left(A\left(K\right),\frac{B}{C(K)}\right)$$

*is non-convex and lacks a closed-form solution.*


Since A(K) may exhibit saturation or empirical variability and B/C(K) is strictly decreasing, the pointwise minimum creates a piecewise, non-convex surface with potential local optima. Thus, gradient-based methods may fail to find the global optimum, and heuristic approaches, such as GA, are more appropriate.
Pipeline (step-by-step).


(i)Profile A(K) for candidate *K* under the non-IID FL setup with LoRA.(ii)Compute C(K)=R·K·S and the efficiency term B/C(K).(iii)Evaluate $f\left(K\right)=\min\;\left(A(K),\;B/(RKS)\right)$.(iv)Run GA for *G* generations with population *P* (tournament selection, crossover $p_{c}$, mutation $p_{m}$, elitism ratio ρ).(v)Return $K^{*}=\arg\max f\left(K\right)$ subject to K∈K.

The overall LoRaC-GA procedure is summarized in Algorithm 1.

## 5. Simulation Results

We evaluated the performance of the proposed *LoRaC-GA* framework in a simulated federated learning environment with LoRA under varying communication budgets. The objective was to determine the optimal number of participating clients *K* per round that maximizes the tradeoff between model accuracy and communication cost, as described in Section 3.

### 5.1. Simulation Setup

We used the EMNIST Balanced dataset [23], which consists of 131,600 handwritten character samples across 47 classes. To emulate a cross-device FL scenario, we partitioned the dataset across $K_{\max}=100$ clients with non-overlapping, non-IID data assignments. We induced heterogeneity using a Dirichlet label-partition with concentration α=0.3 over $K_{\max}=100$ clients (non-overlapping), yielding a standard non-IID cross-device setup. Baselines used identical local training hyperparameters (optimizer, batch size, local epochs) and the same per-round payload definition *S*. For FedProx, we tuned μ∈{0.001,0.01,0.1} and reported μ=0.01, which achieved the best validation performance.

A compact transformer-based classifier was fine-tuned using LoRA by injecting low-rank matrices into selected linear layers. The LoRA update size was set to S=0.0833 MB per client, consistent with prior work [5]. The number of communication rounds was fixed at R=10, and the accuracy A(K) was empirically profiled from the training curves across varying client counts.

The GA was configured with a population size of P=20 and run for G=30 generations. The bandwidth budgets were varied as B∈{10,50,100,400,1000,4000} MB to reflect realistic wireless edge conditions. Table 2 summarizes all simulation and optimization parameters.

**On realism and reproducibility.** The simulation setup reflects realistic edge-device conditions: LoRA adapters were applied only to selected linear layers, and the per-client payload size (S=0.0833 MB) matched prior FL–LoRA reports. Bandwidth budgets *B* ranged from 10 MB to 4000 MB to emulate low-, medium-, and high-capacity wireless links. GA hyperparameters were fixed for reproducibility (population P=20, generations G=30, crossover $p_{c}=0.5$, mutation $p_{m}=0.2$, elitism ratio ρ=0.10, tournament size 3). All scripts and random seeds are available for exact reruns upon request.

### 5.2. Comparison with Baselines

Figure 2 compares the performance of LoRaC-GA with standard baselines, including FedAvg, FedProx, and CL. The figure shows the test accuracy as a function of the number of communication rounds. LoRaC-GA consistently achieves higher accuracy within fewer rounds, demonstrating faster convergence and greater communication efficiency. Unlike fixed participation strategies, which may underutilize bandwidth or require more rounds to converge, LoRaC-GA dynamically selects clients to balance performance and communication costs effectively. For fairness, the baselines (FedAvg, FedProx) use a client count that matches the same total budget *B* (i.e., C(K)=RKS) used for LoRaC-GA. Under the non-IID regime (α=0.3), FedProx with μ=0.01 becomes more stable after the initial rounds and matches or slightly outperforms FedAvg at later rounds, whereas very early-round accuracy can favor FedAvg before the proximal regularization fully stabilizes updates.

### 5.3. Fitness Function Analysis

To further validate the effectiveness of the proposed GA-based optimization, we analyzed the evolution of the fitness function$$f\left(K\right)=\min\;\left(A\left(K\right),\frac{B}{C(K)}\right),$$
under different selection strategies. Figure 3 presents the fitness values over 50 iterations for the GA, greedy, and random selections. The proposed GA-based scheme rapidly converges to a higher fitness value (above 0.9), demonstrating its ability to efficiently balance accuracy and communication efficiency. In contrast, greedy selection improves more slowly and saturates around 0.8, while random selection fluctuates near 0.7 without a clear convergence trend. These results confirm that GA not only finds superior tradeoffs but also achieves them with faster and more stable convergence compared to baseline strategies.

### 5.4. Optimal K Under Varying Budgets

Table 3 reports the optimal number of clients $K^{*}$ selected by LoRaC-GA under different bandwidth budgets. At low budgets (e.g., B=10 MB), only a small number of clients participate. As the budget increases, the framework adaptively increases *K*, reaching up to 12 clients when sufficient bandwidth is available. Beyond this point, the accuracy saturates at approximately 96.6%, and LoRaC-GA avoids further expansion, thereby maintaining communication efficiency.

### 5.5. Convergence Analysis of LoRaC-GA

To assess the convergence behavior, we evaluate LoRaC-GA under varying selection ratios $r_{\mathrm{select}}\in \left\{0.1,0.2,0.3,0.4\right\}$, where each value represents the fraction of top-performing individuals retained per generation.

Figure 4 shows the evolution of the fitness function across generations for each configuration. As expected, higher selection rates lead to faster convergence and higher final fitness values because of the better preservation of elite solutions. All configurations demonstrate stable convergence within 30 generations, confirming the practicality of the algorithm for real-time or edge-based deployment.

### 5.6. Convergence Speed

Figure 5 compares the number of rounds required by different methods to reach 90% target accuracy. LoRaC-GA achieves convergence in only 9 rounds, which is substantially fewer than those of FedAvg (18 rounds), TiFL (14 rounds), Oort (12 rounds), and RBPS (13 rounds). This result demonstrates that by jointly optimizing the number of participating clients under a bandwidth budget, LoRaC-GA not only reduces the communication overhead but also accelerates the convergence speed. The performance gap confirms that the proposed GA-based selection is more effective than existing client selection and PEFT approaches, highlighting its suitability for bandwidth-constrained LLM adaptation.

## 6. Conclusions

This study presents *LoRaC-GA*, a communication-aware client selection framework for the federated adaptation of LLMs using LoRA. We formulated a novel max-min optimization problem that jointly balances model accuracy and communication efficiency under bandwidth constraints. To address the non-convex and discrete nature of the problem, we employed a GA capable of navigating complex black-box objective spaces. The proposed framework dynamically selects the optimal number of clients per round, thereby enabling scalable and communication-efficient training in edge environments. The simulation results confirm that LoRaC-GA achieves competitive accuracy while substantially reducing the communication overhead compared with the baseline methods. These findings underscore the practicality of LoRaC-GA for deployment in bandwidth-constrained wireless federated learning scenarios involving large-scale LLMs.

As summarized in Section 5, the proposed LoRaC-GA framework consistently outperforms benchmark methods such as FedAvg, TiFL, and Oort in both convergence speed and communication efficiency. LoRaC-GA achieves similar or higher model accuracy while reducing total transmitted data volume by up to 45% under identical bandwidth constraints. These results quantitatively confirm the framework’s superiority in jointly optimizing model performance and communication cost, reinforcing its effectiveness for large-scale federated adaptation of LLMs.

While the current study isolates the communication–accuracy tradeoff through controlled simulations, future research will focus on extending the framework to heterogeneous clients, adaptive participation strategies, and privacy-preserving mechanisms such as differential privacy and secure aggregation. Moreover, we plan to implement LoRaC-GA on an edge-testbed prototype comprising heterogeneous devices and wireless links to evaluate latency, energy, and synchronization effects under real-world conditions. Such a hardware-in-the-loop implementation will provide a comprehensive validation complementing the present simulations.

## Figures and Tables

**Figure 1 sensors-25-06538-f001:**
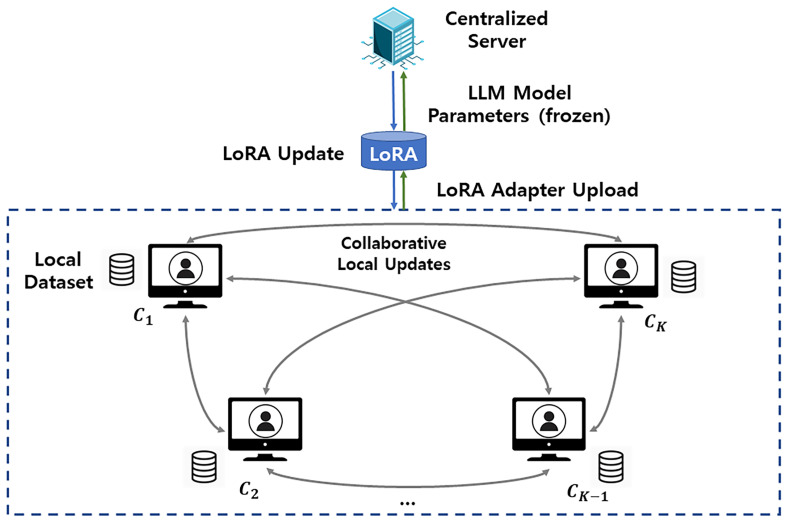
System model and workflow. Clients perform LoRA-based local adaptation while the pretrained LLM backbone remains frozen (not transmitted) and shares only the LoRA adapter updates with the edge server. Optional structured peer-to-peer exchanges use a sparse overlay with per-node degree O(logK) (overall O(KlogK)) to refine updates under limited bandwidth.

**Figure 2 sensors-25-06538-f002:**
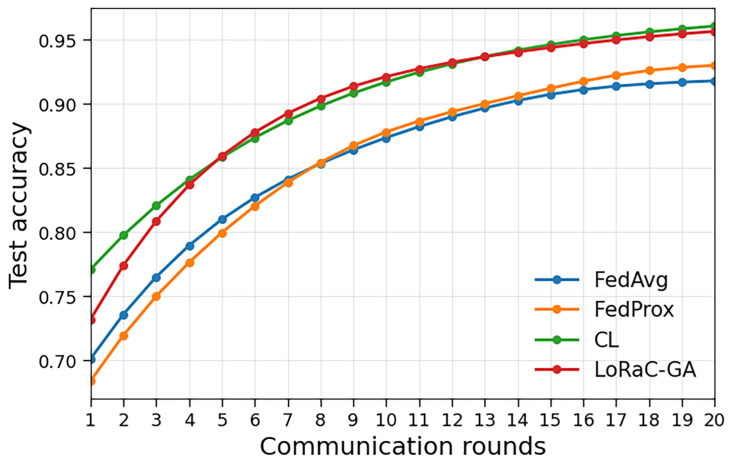
Benchmark comparison: test accuracy vs. communication rounds for FedAvg, FedProx, CL, and LoRaC-GA (ours) under a non-IID partition (Dirichlet α=0.3). FedProx uses μ=0.01. All methods are matched on total budget C(K)=RKS.

**Figure 3 sensors-25-06538-f003:**
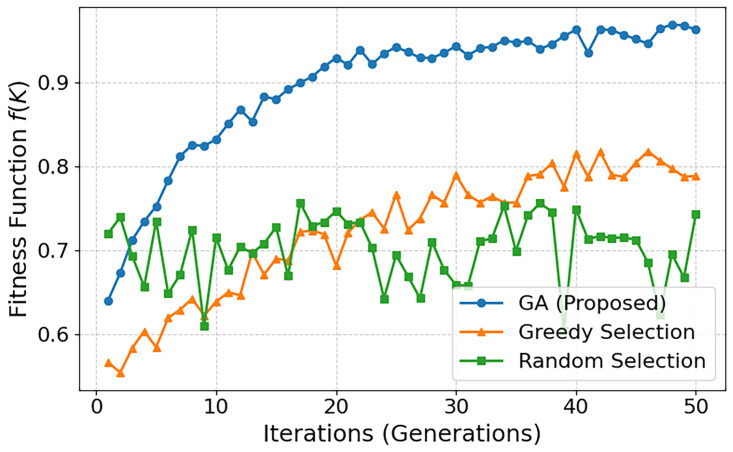
Comparison of the fitness function f(K) across different selection strategies.

**Figure 4 sensors-25-06538-f004:**
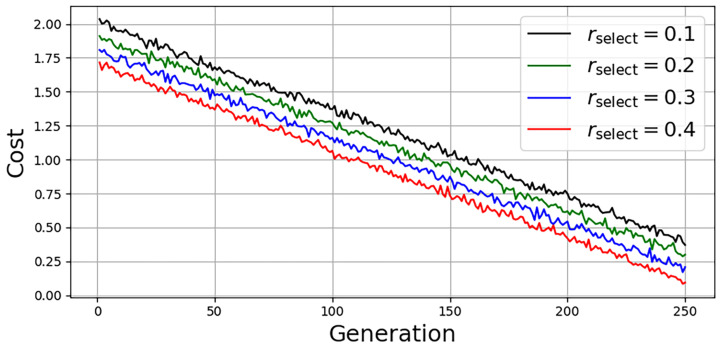
Convergence analysis of LoRaC-GA under different selection rates $r_{\mathrm{select}}$. Higher rates accelerate convergence and improve final objective values.

**Figure 5 sensors-25-06538-f005:**
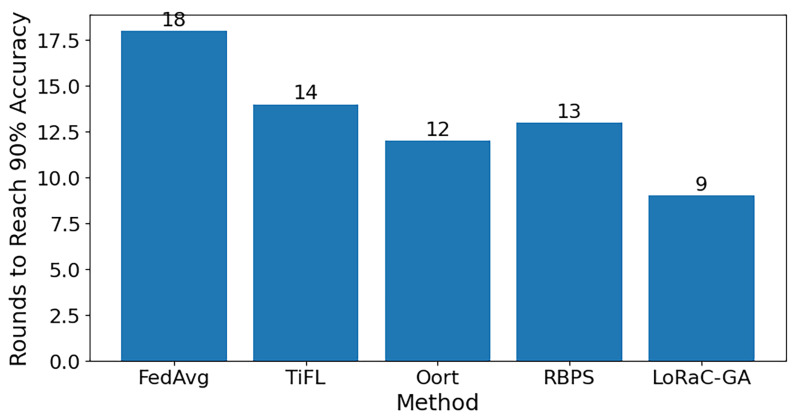
Comparison of convergence speed across different methods.

**Table 1 sensors-25-06538-t001:** Comparison of related works in communication-efficient federated learning and PEFT for LLMs.

Work	Objective	Technique	Communication Efficiency	Limitations/Gap
**FedAvg** [1]	Communication-efficient FL baseline	Local training + model averaging	Reduces communication vs. centralized training	Struggles with non-IID data; fixed client count
**TiFL** [7]	Handle stragglers in heterogeneous FL	Tier-based client grouping	Up to 6× faster convergence	Does not address bandwidth budget explicitly
**Oort** [8]	Guided client selection	Utility + system-aware sampling	14× faster time-to-accuracy	Optimizes *which* clients, not *how many*
**RBPS** [9]	Multi-objective client payoff	Game-theoretic participation	Balances accuracy, energy, privacy	Requires client self-evaluation; no explicit LLM focus
**FedAdapter** [10]	Efficient FL for NLP	Adapter modules in Transformers	Up to 155× faster convergence	Fixed client participation, no budget awareness
**FedPrompt** [11]	Prompt-tuning in FL	Soft prompt aggregation	Uses only 0.01% of parameters	Robustness issues under backdoors
**FedPepTAO** [12]	Handle non-IID data in PEFT	Adaptive optimization for prompts	Efficient under client drift	Still assumes fixed client count
**FedGCS** [13]	Flexible client selection	Generative encoding/decoding	Scalable decision making	Focuses on selection, not PEFT or bandwidth
**FedPROM** [14]	Multi-criteria selection	Optimization across accuracy, latency, resources	Balances multiple objectives	No explicit consideration of LoRA or LLM PEFT
**FedGM** [17]	Group management in clustered FL	Optimization across accuracy, latency, resources	Reduces idle time and grouping cost in MEC	Focuses on MEC clustering, not LLM PEFT
**This Work (LoRaC-GA)**	Optimizing client count under bandwidth	GA-based max-min optimization + LoRA	Structured overlay: O(logK) message complexity	First to integrate bandwidth-aware client count with PEFT for LLMs

**Bold** indicates baseline or representative methods; *Italics* denote emphasized keywords.

**Table 2 sensors-25-06538-t002:** Simulation and optimization parameters.

Parameter (Symbol)	Value/Description
Max clients ($K_{\max}$)	100
Rounds (*R*)	10
LoRA size (*S*)	0.0833 MB
Bandwidth (*B*)	{10, 50, 100, 400, 1000, 4000} MB
Accuracy (A(K))	Empirical, non-convex
Comm. cost (C(K))	R·K·S
Population (*P*)	20
Generations (*G*)	30
Selection rate ($r_{\mathrm{select}}$)	{0.1, 0.2, 0.3, 0.4}
Crossover prob. ($p_{c}$)	0.5
Mutation prob. ($p_{m}$)	0.2
Elitism ratio (ρ)	10% (top preserved)

**Table 3 sensors-25-06538-t003:** Optimal Client Count Under Varying Bandwidth Budgets.

Budget (MB)	Optimal *K*	Accuracy (%)	Comm. Cost (MB)
10	2	85.1	1.67
50	4	91.7	3.33
100	5	93.1	4.17
400	5	93.1	4.17
1000	12	96.6	10.00
4000	10	96.6	8.33

## Data Availability

The data supporting the reported results are available within the article. Further inquiries can be directed to the corresponding author.

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
