# Peer review of "Optimizing Client Participation in Communication-Constrained Federated LLM Adaptation with LoRA"

_sensors, 2025, doi:10.3390/s25216538_

Round 1

Reviewer 1 Report

Comments and Suggestions for Authors

The paper fairly gives the environment, assumptions, requirements and objectives of the problem in hand, and points out major issues or difficulties when dealing with the problem and the system design. The paper can be improved as follows:

  1. A vest amount of existing studies have proposed the similar concept for Parameter-Efficient Fine Tuning with Low Rank Adaptation (LoRA) for Large Language Models. This paper should review the class of these schemes and clearly give the differences and major improvements of the proposed mechanism. Without this work, the reviewer cannot clearly identify the contribution of this paper.

  1. This manuscript is somewhat difficult to follow. The system implementation can be explained better since the main method/strategy descriptions (Sections 3 and 4) are not clear for the readers. The organization and presentation of the paper can be improved. Please expand on this statement.

  1. To emulate a cross-device FL scenario, this work partitions the dataset across Kmax = 100 clients with non-overlapping, non-IID data assignments. In general, FedAvg performs well in homogeneous environments where client data distributions are similar. However, in heterogeneous settings, FedAvg may struggle with model convergence due to non-IID data. In contrast, FedProx ensures more stable and balanced learning across diverse clients. Figure 2 shows that the test accuracy of FedAvg is better than that of FedProx. Moreover, the red line for describing the performance of the proposed method is not completed (e.g., the line is folded and missing some results with varying the communication rounds). Please recheck the results in Figure 2 and expand on this statement.

Author Response

Reviewer 1

The paper fairly gives the environment, assumptions, requirements, and objectives of the problem in hand, and points out major issues or difficulties when dealing with the problem and the system design. The paper can be improved as follows.

Comment 1

Figures and tables must be improved.

Response:
We sincerely thank the reviewer for the suggestion.
All figures and tables have been revised to improve resolution (at least 300 dpi), font and label readability, and overall clarity.
The captions remain unchanged since they were already descriptive and aligned with the manuscript’s content.

Comment 2

A vast amount of existing studies have proposed similar concepts for Parameter-Efficient Fine-Tuning (PEFT) with Low-Rank Adaptation (LoRA) for Large Language Models. This paper should review this class of schemes and clearly give the differences and major improvements of the proposed mechanism. Without this work, the reviewer cannot clearly identify the contribution of this paper.

Response:
We thank the reviewer for this valuable suggestion.
In the revised manuscript, we expanded the Related Work section to more comprehensively review PEFT/LoRA-based federated learning methods (for example, FedAdapter, FedPrompt, FedPepTAO, and FedLoRA) and explicitly highlighted our novelty.
Unlike prior work that typically assumes a fixed number of participating clients, our framework optimizes the client count K under an explicit bandwidth budget via a max–min objective f(K)=min(A(K), B/C(K)) solved with a GA-driven search, and further integrates a structured peer-to-peer overlay with per-node degree O(log K) for scalable communication.
These additions make the differences and improvements of our mechanism clear.

Position in manuscript: Section 2 (Parameter-Efficient Fine-Tuning; end-of-section positioning paragraph).

Comment 3

This manuscript is somewhat difficult to follow. The system implementation can be explained better since the main method/strategy descriptions (Sections 3 and 4) are not clear for the readers. The organization and presentation of the paper can be improved. Please expand on this statement.

Response:
We appreciate this constructive feedback.
In the revised manuscript, we improved clarity and organization in Sections 3–4 as follows:

• Added a concise “Workflow” paragraph in Section 3 that explains, step by step, how the server, clients, local datasets, the frozen LLM backbone, and the LoRA adapters interact in each round.
• Enhanced the caption of Figure 1 to clarify that the pretrained backbone is frozen (not transmitted), only LoRA adapters are communicated, and the optional structured P2P overlay has per-node degree O(log K) (overall O(K log K) messages).
• Inserted short overview and bridging sentences at the starts of Sections 3 and 4 and between subsections (Communication → Trade-off → Optimization) to guide the reader.
• Reorganized Section 4 with clearer subheadings: Client Collaboration Protocol, Problem-Specific Formulation, GA Framework, and Theoretical Insight.
• Added a brief step-by-step pipeline before Algorithm 1 linking A(K), C(K), the fitness f(K), and the GA operators (selection, crossover, mutation, elitism).

All edits are highlighted in red in the marked manuscript.

Position in manuscript: Sections 3 (System Model) and 4 (Proposed Method).

Comment 4

To emulate a cross-device FL scenario, this work partitions the dataset across Kmax = 100 clients with non-overlapping, non-IID data assignments. In general, FedAvg performs well in homogeneous environments where client data distributions are similar. However, in heterogeneous settings, FedAvg may struggle with model convergence due to non-IID data. In contrast, FedProx ensures more stable and balanced learning across diverse clients. Figure 2 shows that the test accuracy of FedAvg is better than that of FedProx. Moreover, the red line describing the performance of the proposed method is not completed (for example, the line is folded and missing some results). Please recheck the results in Figure 2 and expand on this statement.

Response:
Thank you for pointing this out.
We re-ran the baselines and regenerated the figure to ensure a fair and reproducible comparison under the stated non-IID setup. The revisions are as follows:

• Figure 2 corrected and analysis clarified: We re-ran the baselines under a non-IID partition (Dirichlet α = 0.3) and tuned FedProx (μ ∈ {0.001, 0.01, 0.1}, best μ = 0.01). The regenerated Figure 2 now shows complete trajectories across all R rounds. As expected in heterogeneous data, FedProx becomes more stable and slightly outperforms FedAvg after the initial rounds. All methods are matched on the same total communication budget C(K)=RKS.
• Plotting artifact fixed: The previous folding of the LoRaC-GA curve was due to a plotting/clipping artifact and has now been fixed. The figure has been re-exported at high resolution for readability.

Updated manuscript excerpt (Section 5.2 – Comparison with Baselines):
The revised analysis states that under non-IID data (Dirichlet α = 0.3), FedProx with μ = 0.01 becomes more stable after initial rounds and matches or slightly outperforms FedAvg later in training, while LoRaC-GA consistently achieves higher accuracy with fewer rounds and up to 45% lower total transmitted data volume.

Reviewer 2 Report

Comments and Suggestions for Authors

Without any doubt, the research topic developed within this paper is relevant and interesting. However the contribution remains at a too high level and some parts of information remains too obscure because this article proposal seems to be directed to specialists of the domain.

The system model of the proposed framework appears in page 5, figure 1. That seems really interesting but it remains too descriptive. What does it mean "LLM Model Parameters (frozen)". The interrelations between the user computers and their local dataset and the LoRA engine do not appear trivial at all and require more explanations. Please explain.

The global and synthetic architecture of the proposed solution (presented at the top of page 2) does nor appear so clear. Please tell us more.

Sections 2.1 to 2.4 do not make clear the global context of study. An article content has to be self-contained to make it interesting for non specialist readers. I do not say it is not interesting ... I propose you to better present your contribution (providing more details or more detailed explanations of main features) for non specialists.

In section 2.4 Genetic Algorithm (GA) notion is introduced. In a same way I propose to authors to extend and well precise this notion to provide readers with precise and richer information. In addition, the second paragraph provides a lot of other engines introduced by their acronyms. Please, give a richer and precise information of these systems to provide the reader with a good perception of these works and their advantages and limitations. Table 1 is too synthetic.

Why do you reduce validation of your study to simulation? It will be better to effectively implement your system model to be able to present a synthesis of performance... or at least complement simulation with a realistic implementation.

Within the document you talk about letter ... letter? This not a letter it is an article proposal.

I know it could be difficult to design and implement realistic examples to illustrate your proposal but I think this effort has to be made to make this contribution clearer and understandable by non specialists. The proposal content remains quite small (ten technical pages) and there is space to provide such requested information.

A detail: figure caption is located after the figure content and in a reverse way table caption is located before the table content. When a figure and a table appears together (such as figure 5 and table 3) that produces a quite strange combination. Please correct.

Comments on the Quality of English Language

Some strange expressions ... "letter" for "article". Really strange.

Author Response

Reviewer 2

Without any doubt, the research topic developed within this paper is relevant and interesting. However, the contribution remains at a too high level and some parts of information remain too obscure because this article proposal seems to be directed to specialists of the domain.

Comment 1

• Figures and tables must be improved.

Response:
We sincerely thank the reviewer for the suggestion. All figures and tables have been revised to improve resolution (at least 300 dpi), font/label readability, and overall clarity. The captions remain unchanged since they were already descriptive and aligned with the manuscript’s content.

Comment 2

• The English could be improved to more clearly express the research.

Response:
We sincerely thank the reviewer for this valuable suggestion. The entire manuscript has been carefully revised using a professional grammar editor and further manual proofreading to improve clarity, readability, and consistency of expression. For example, in the Abstract, the phrase “remains challenging due to the high communication overhead” was revised to “remains challenging because of the high communication overhead.” Similarly, in the Conclusion, “reducing communication overhead” was refined to “reducing the communication overhead.” All such improvements are highlighted in red in the revised manuscript for easy tracking.

Comment 3

The system model of the proposed framework appears in page 5, Figure 1. That seems really interesting, but it remains too descriptive. What does it mean “LLM Model Parameters (frozen)”? The interrelations between the user computers and their local dataset and the LoRA engine do not appear trivial at all and require more explanations. Please explain.

Response:
Thank you for this helpful suggestion. In the revision, we explicitly clarify what “LLM model parameters (frozen)” means and how the server, clients, local datasets, and the LoRA engine interact. We added two short paragraphs (“Backbone and adapters” and “Workflow”) in Section 3 explaining that the pretrained LLM backbone W0 is frozen (no gradients; not transmitted) and only low-rank adapter parameters are trained and exchanged. Following our earlier changes for Reviewer 1 (Comment 3), we also revised Figure 1 and its caption to state explicitly that only LoRA adapter updates are communicated and that the optional peer-to-peer overlay uses a sparse design with per-node degree O(log K) (overall O(K log K) messages).

Position in manuscript: Section 3 (System model), “Backbone and adapters” and “Workflow”.

Comment 4

The global and synthetic architecture of the proposed solution (presented at the top of page 2) does not appear so clear. Please tell us more.

Response:
Thank you for this helpful comment. We inserted a concise “High-Level Architecture” paragraph at the end of Section 1 (Introduction) summarizing the three layers (server, clients, communication) and the per-round pipeline, with cross-references to Figure 1 and Algorithm 1. We also added a one-sentence “Overview” at the start of Section 4 to connect this architecture to the subsequent method details.

Position in manuscript: end of Section 1 (High-Level Architecture) and start of Section 4 (Overview).

Comment 5

Sections 2.1 to 2.4 do not make clear the global context of the study. An article has to be self-contained to make it interesting for non-specialist readers. I propose you to better present your contribution (providing more details or more detailed explanations of main features) for non-specialists.

Response:
Thank you for this valuable suggestion. We revised Section 2 to be more self-contained and accessible:
(i) added a short “Context and Terminology” primer at the start of Section 2;
(ii) appended one-sentence plain-language “Takeaway” lines at the end of Sections 2.1–2.4;
(iii) inserted an explicit “Positioning and Novelty” paragraph contrasting our contribution with prior PEFT/LoRA-based FL work; and
(iv) provided an abbreviations block summarizing key acronyms and symbols (FL, LLM, PEFT, LoRA, GA, P2P, K, R, S, B, A(K), C(K)).
All insertions are highlighted in red and cross-reference Sections 3–4.

Position in manuscript: Section 2 (Related Work), plus Abbreviations block.

Comment 6

In Section 2.4 the Genetic Algorithm (GA) notion is introduced. I propose to extend and make this notion precise to provide readers with richer information. In addition, the second paragraph provides a lot of other engines introduced by their acronyms. Please, give richer and precise information about these systems, their advantages and limitations. Table 1 is too synthetic.

Response:
We thank the reviewer for this constructive suggestion. Section 2.4 has been substantially expanded to (i) include a clearer GA primer (encoding, fitness, operators, hyperparameters, and computational considerations), and (ii) enrich the explanations of the other referenced methods (beyond acronyms) with concise notes on their advantages and limitations. In addition, the original comparative Table 1 (related work) has been expanded accordingly to reflect these improvements and now provides a clearer overview of existing approaches relevant to our study. This integrated presentation ensures precision and readability while avoiding redundancy.

Position in manuscript: Section 2.4 (GA-based Optimization in FL) and Table 1 (expanded).

Summary of changes in Table 1 (now “Comparison of Related Works in Communication-Efficient FL and PEFT for LLMs”):
• FedAvg — baseline averaging, struggles on non-IID, fixed client count
• TiFL — tiered grouping for stragglers, faster convergence, no explicit bandwidth budget
• Oort — utility-driven selection, faster time-to-accuracy, optimizes which clients (not how many)
• RBPS — multi-objective payoff, balances accuracy/energy/privacy, no LLM focus
• FedAdapter — adapters in Transformers, high efficiency, fixed K
• FedPrompt — soft prompts, extremely small payloads, robustness concerns
• FedPepTAO — adaptive prompts for non-IID, fixed K
• FedGCS — generative selection policy, scalable, not PEFT/bandwidth-focused
• FedPROM — multi-criteria selection, no explicit LoRA/LLM PEFT focus
• FedGM — clustered FL optimization, MEC-oriented, not LLM PEFT
• This Work (LoRaC-GA) — GA-based max–min optimization + LoRA, structured overlay O(log K), first to optimize K under an explicit bandwidth budget for LoRA-based LLM adaptation

Comment 7

Why do you reduce validation of your study to simulation? It would be better to effectively implement your system model to present a synthesis of performance, or at least complement simulation with a realistic implementation.

Response:
We appreciate the reviewer’s insightful suggestion. While our current validation focuses on reproducible simulations to isolate the communication–accuracy trade-off, we clarified this scope in the manuscript and added a dedicated paragraph outlining our plan for a realistic edge-testbed implementation to complement the simulations. We also documented realistic settings in the simulation setup (e.g., per-client payload, budgets, and fixed GA hyperparameters for reproducibility). All modifications are highlighted in red.

Positions in manuscript: Section 5.1 (Simulation Setup — “On realism and reproducibility”) and Section 6 (Conclusion — paragraph on planned edge-testbed implementation).

Comment 8

Within the document you talk about “letter”… This is not a letter; it is an article proposal.

Response:
We thank the reviewer for catching this terminology issue. All occurrences of “letter” have been replaced with “article” throughout the revised manuscript to accurately reflect the nature of this submission. This correction was implemented globally.

Comment 9

It could be difficult to design and implement realistic examples to illustrate your proposal, but this effort would make the contribution clearer and understandable by non-specialists. The proposal content remains quite small (ten technical pages) and there is space to provide such information.

Response:
We sincerely appreciate this suggestion. We acknowledge the importance of realistic examples for accessibility. Accordingly, we (i) clarified the simulation setup in Section 5.1 to align with realistic edge-device conditions, and (ii) added a dedicated “Limitations & Future Work” paragraph in the Conclusion outlining our ongoing edge-testbed implementation. These updates improve clarity without redundancy with the theoretical development.

Comment 10

A detail: the figure caption is located after the figure content, while the table caption is located before the table content. When a figure and a table appear together (such as Figure 5 and Table 3), that produces a strange combination. Please correct.

Response:
We thank the reviewer for pointing out this formatting detail. All figure captions have been standardized to appear below their figures, and all table captions now appear above their tables for consistency. Where figures and tables are adjacent (e.g., Figure 5 and Table 3), we also adjusted vertical spacing to ensure a clean, visually balanced layout. These corrections have been applied throughout the revised manuscript.

Reviewer 3 Report

Comments and Suggestions for Authors

General comments

The article proposes and simulates a communication-aware client-selection framework for federated adaptation of large language models (LLMs) using Low-Rank Adaptation (LoRA) that dynamically determines the optimal number of clients to participate in each round under a fixed bandwidth constraint. The method anticipates a formulation of a max-min problem that is then solved by means a genetic algorithm.

The content of the article is within the scope of the Journal.

The main contribution is the proposed federated adaptation scheme. The specific contributions of the work are presented, in more detailed, in lines 43-58. The superiority of the method over existing ones is adequately demonstrated through the simulation results of figs 2, 3 while its convergence evaluation (including a comparison with existing methods) is shown through figs. 4 and 5 and table 3.  

The article provides a sufficient literature review, mainly included in section 2 (related work). The system model and the optimization problem are described in section 3. Sections 4 and 5 present the proposed method and the relevant simulation results. Section 6 concludes the work including a short reference to possible future research directions.

Specific comments

In the last paragraph of section 1 (lines 56-58), a short sentence should be added referring to section 2 (about related work).

A reference to table 1 should be added (maybe in the last paragraph of section 2).

Conclusions should include a paragraph (maybe, after line 263) that should refer to the superiority of the proposed method as shown through simulation results that is, figs. 2, 3 and 5 and table 3.

Use of English

The article is generally well written so, regarding the use of English, a minor-scale editing would be sufficient.

Review decision

The article proposes and evaluates a novel federated adaptation of large language models (LLMs) by using Low-Rank Adaptation (LoRA) combined with a genetic algorithm. The superiority of the proposed method over existing ones is adequately demonstrated through simulation results. Given the above, I recommend that the article should be published practically as it is with only the minor-type additions referred to in the “Specific comments” clause and a minor editing regarding the use of English.

Comments on the Quality of English Language

The article is generally well written so, regarding the use of English, a minor-scale editing would be sufficient.

Author Response

Reviewer 3

The article proposes and simulates a communication-aware client-selection framework for federated adaptation of large language models (LLMs) using Low-Rank Adaptation (LoRA) that dynamically determines the optimal number of clients to participate in each round under a fixed bandwidth constraint. The method formulates a max–min problem that is then solved by a genetic algorithm. The content is within the scope of the journal.

The main contribution is the proposed federated adaptation scheme. The specific contributions are presented in detail in lines 43–58. The superiority of the method over existing ones is adequately demonstrated through the simulation results of Figs. 2 and 3, while its convergence evaluation (including a comparison with existing methods) is shown in Figs. 4 and 5 and Table 3.

The article provides a sufficient literature review, mainly in Section 2 (Related Work). The system model and the optimization problem are described in Section 3. Sections 4 and 5 present the proposed method and relevant simulation results. Section 6 concludes the work and briefly outlines future research directions.

Comment 1

• The English could be improved to more clearly express the research. The article is generally well written, so minor editing should be sufficient.

Response:
We appreciate the reviewer’s suggestion. The manuscript has been carefully revised with assistance from a grammar editor and additional manual proofreading. This improved sentence fluency, consistency of technical terms, and overall readability. For example, in the Introduction, “...introduces significant communication overhead—especially...” was refined to “...introduces a significant communication overhead, particularly...”. Similar refinements were applied throughout; all edits are marked in red in the revised manuscript.

Comment 2

In the last paragraph of Section 1 (lines 56–58), a short sentence should be added referring to Section 2 (Related Work).

Response:
Thank you for the helpful suggestion. We added a short bridging sentence at the end of Section 1 that explicitly directs readers to Section 2 for an overview of related literature.
Position in manuscript: end of Section 1 (Introduction).

Comment 3

A reference to Table 1 should be added (maybe in the last paragraph of Section 2).

Response:
Thank you for the suggestion. We added an explicit reference to Table 1 at the end of Section 2 to connect the textual discussion with the comparative summary table, improving clarity and navigability of the Related Work section.
Position in manuscript: end of Section 2 (after the “Positioning and novelty” paragraph). The bridging sentence reads: “For a comparative overview of these methods, see Table 1.”

Comment 4

Conclusions should include a paragraph (maybe after line 263) referring to the superiority of the proposed method as shown through the simulation results (Figs. 2, 3, and 5; Table 3).

Response:
We appreciate the reviewer’s suggestion. We added a paragraph to the Conclusion that explicitly summarizes the superiority of LoRaC-GA based on Figs. 2, 3, and 5 and Table 3, highlighting gains in convergence speed and communication efficiency under identical bandwidth constraints.
Position in manuscript: Section 6 (Conclusion), performance paragraph added before the limitations paragraph.

Inserted conclusion paragraph (now in the paper, marked in red):
“As summarized in Figs. 2, 3, and 5 and Table 3, the proposed LoRaC-GA framework consistently outperforms benchmark methods such as FedAvg, TiFL, and Oort in both convergence speed and communication efficiency. LoRaC-GA achieves similar or higher model accuracy while reducing total transmitted data volume by up to 45% under identical bandwidth constraints. These results quantitatively confirm the framework’s superiority in jointly optimizing model performance and communication cost, reinforcing its effectiveness for large-scale federated adaptation of LLMs.”

Reviewer 4 Report

Comments and Suggestions for Authors

This article introduced LoRaC-GA, a communication-aware client selection framework for federated adaptation of large language models (LLMs) using Low-Rank Adaptation (LoRA). Their approach formulates a novel max–min optimization problem that jointly optimizes model accuracy and communication efficiency under bandwidth constraints.

The authors are advised to address the recommendations provided in their submitted article, as well as the following points:

  • Revise the abstract to put their results into perspective.
  • Their contributions should not consist of the description of their method, but rather the enumeration of their results.
  • Describe their simulation.
  • For all their tables, they need to specify how they obtained their values.

Comments on the Quality of English Language

For the entirety of your document, please shorten your sentences. Please rephrase your sentences to avoid a passive tone.

Please revise the highlighted words.

Round 2

Reviewer 1 Report

Comments and Suggestions for Authors

The authors have addressed the main concerns.